# Exploring Alzheimer’s Disease Molecular Variability via Calculation of Personalized Transcriptional Signatures

**DOI:** 10.3390/biom10040503

**Published:** 2020-03-26

**Authors:** Hila Dagan, Efrat Flashner-Abramson, Swetha Vasudevan, Maria R. Jubran, Ehud Cohen, Nataly Kravchenko-Balasha

**Affiliations:** 1The Rachel and Selim Benin School of Computer Science and Engineering, Hebrew University, Jerusalem 9190416, Israel; Hila.dagan@mail.huji.ac.il; 2Department for Bio-Medical Research, Faculty of Dental Medicine, Hebrew University of Jerusalem, Jerusalem 91120, Israel; efrat.flashner@mail.huji.ac.il (E.F.-A.); swetha.vasudevan@mail.huji.ac.il (S.V.); maria.jubran@mail.huji.ac.il (M.R.J.); 3Department of Biochemistry and Molecular Biology, The Institute for Medical Research Israel—Canada, The Hebrew University School of Medicine, Jerusalem 9112102, Israel; ehudc@ekmd.huji.ac.il

**Keywords:** Alzheimer’s disease, information theory, altered transcriptional network structure, surprisal analysis, patient-specific transcriptional signatures

## Abstract

Despite huge investments and major efforts to develop remedies for Alzheimer’s disease (AD) in the past decades, AD remains incurable. While evidence for molecular and phenotypic variability in AD have been accumulating, AD research still heavily relies on the search for AD-specific genetic/protein biomarkers that are expected to exhibit repetitive patterns throughout all patients. Thus, the classification of AD patients to different categories is expected to set the basis for the development of therapies that will be beneficial for subpopulations of patients. Here we explore the molecular heterogeneity among a large cohort of AD and non-demented brain samples, aiming to address the question whether AD-specific molecular biomarkers can progress our understanding of the disease and advance the development of anti-AD therapeutics. We studied 951 brain samples, obtained from up to 17 brain regions of 85 AD patients and 22 non-demented subjects. Utilizing an information-theoretic approach, we deciphered the brain sample-specific structures of altered transcriptional networks. Our in-depth analysis revealed that 7 subnetworks were repetitive in the 737 diseased and 214 non-demented brain samples. Each sample was characterized by a subset consisting of ~1–3 subnetworks out of 7, generating 52 distinct altered transcriptional signatures that characterized the 951 samples. We show that 30 different altered transcriptional signatures characterized solely AD samples and were not found in any of the non-demented samples. In contrast, the rest of the signatures characterized different subsets of sample types, demonstrating the high molecular variability and complexity of gene expression in AD. Importantly, different AD patients exhibiting similar expression levels of AD biomarkers harbored distinct altered transcriptional networks. Our results emphasize the need to expand the biomarker-based stratification to patient-specific transcriptional signature identification for improved AD diagnosis and for the development of subclass-specific future treatment.

## 1. Background 

Alzheimer’s disease (AD) is the most common cause of dementia, characterized by progressive cognitive decline and neurodegeneration. AD is defined pathologically by the presence of senile plaques and neurofibrillary tangles (NFTs), particularly in the hippocampus and neocortex [1]. Definitive AD pathology can only be determined by autopsy, since this neurologic manifestation is not readily perceptible using current diagnostic technologies [2]. Nevertheless, research laboratories have been investing large efforts to characterize molecular alterations in AD samples, aiming to deepen our understanding of the disease, and to allow for better diagnosis and stratification of AD patients once brain biopsies will become available.

According to the amyloid hypothesis [3], AD develops as a result of the hyper activation of two proteolytic entities, the β and γ secretases, which both digest the amyloid precursor protein (APP). This dual digestion results in increased production of the family of aggregative amyloid β (Aβ) peptides, which in turn cause neuronal death and underlie the development of AD. Nevertheless, a careful analysis of familial AD (fAD)-causing mutations in the sequence of presenilin 1 (PS1), an aspartic protease which possesses the activity of the γ secretase complex, unveils that many fAD-causing mutations lead to loss of PS1 function [4,5], thereby contradicting the amyloid hypothesis. 

In addition, a comparison of Aβ production levels in brains of individuals who either suffered from fAD, sporadic AD (sAD) or were not demented, indicated that sAD patients and non-demented individuals show no significant differences in Aβ production levels [6]. 

Moreover, different mutations in the sequence of PS1 resulted in dissimilar effects, as some mutations increased and some decreased the levels of Aβ production [6]. 

Further complexity was recently demonstrated by a study showing intricate interactions between APOE (apolipoprotein E, a protein which is involved in Aβ metabolism) genotypes and other common genetic variants associated with AD. These interactions affect the absolute risk to develop the disorder and the age at onset [7]. 

Together, these observations show that the amyloid hypothesis cannot explain all AD cases and strongly suggest that distinct mechanisms underlie the manifestation of this devastating disease. The existence of multiple mechanisms that underlie AD development limits our ability to perform accurate patient diagnosis and develop efficient treatments for AD [8,9]. 

This understanding has spurred the implementation of high-throughput methods (i.e., genomics, transcriptomics, etc.) and the invention of novel analytical tools for the characterization and diagnosis of AD subtypes. 

Recent advances in the development of quantitative tools for AD characterization include multivariate statistical methods, such as clustering methods, principal component analysis [10,11,12], Bayesian methods [13], and machine learning [14]. These methods usually detect dominant, statistically significant groups of co-varying transcripts and proteins, which appear in large numbers of tested brains/brain regions [12]. Hence, uncommon transcript expression patterns within the networks and rare altered network structures that do not include any of the prevailing groups of co-varying transcripts/proteins may be overlooked. 

In clinics, pathologies bearing comparable mutations or biomarker expression levels would be classified as similar. However, when dealing with complex multifactorial pathologies similar biomarker expression levels in different patients may stem from different altered molecular processes [15], possibly necessitating distinct, patient-specific diagnosis and treatment. This is specifically problematic in the case of AD, which appears to be a syndrome rather than a single disease [16]. 

We explore the molecular data space of AD using information theory, aiming to find a way to classify AD patients not only based on biomarker expression levels, but rather based on their complete patient-specific transcriptional network structure. 

A comprehensive classification approach that will enable improved diagnosis and patient stratification may significantly advance the development of AD treatments. Without proper classification of patients, clinical trials may fail to recognize therapies that are beneficial to a sub-group of patients, as the effect may be masked by non-responders included in the trial. 

To identify patient-specific altered transcriptional networks we employ a thermodynamic-based information theoretic method named surprisal analysis (SA). SA determines patterns of altered molecular expression levels in a population of samples, and thereby identifies transcriptional subnetworks that repeat themselves throughout the population [15,17]. A unique attribute of SA is that each transcript can be influenced by several different subnetworks, in line with the non-linearity of biological networks [15,17]. Several distinct subnetworks may operate in each AD/non-demented brain tissue, together constituting a unique altered transcriptional network, or sample-specific transcriptional signature. See the Results and Methods sections below, and [15,17] for more details. 

We have recently demonstrated that similar to systems in chemistry and physics, the interpretation of molecular alterations based on physico-chemical rules [18], e.g., through identification of the altered subnetworks that deviate the system from the steady state, allows the prediction and rational manipulation of biological phenotypes. Examples include predictions of spatial distributions of aggressive brain tumor cells [19], direction of cell-cell movement [20], response of cells to drug treatment [21], and predictions of patient-specific cancer drug combinations [17].

We study herein a large transcriptomic dataset consisting of 737 postmortem brain samples obtained from up to 17 brain regions of 85 sporadic AD patients. Additionally, the dataset contained 214 control postmortem brain samples obtained from up to 17 brain regions of 22 elderly non-demented brains. The dataset was generated and analyzed by Wang et al. [12], who looked for differences and similarities between the various AD brain regions. The authors showed that gene expression alterations in each brain region can be clustered into dozens and sometimes even hundreds of different co-expression gene modules [12]. Thereafter, different AD brain regions were compared in order to find overlapping or highly correlated biological modules. 

We wished to complement the study by Wang et al. [12] by looking at the gene expression alterations in the dataset from a different viewpoint: instead of searching for overlapping/highly correlated molecular aberrations in different brain regions, we examine each brain sample and each patient individually. We decode the patient-specific molecular network reorganization events that occurred in each individual brain sample, namely the patient-specific altered transcriptional signature. We show that 30 distinct altered transcriptional signatures characterize solely AD samples and 22 additional altered transcriptional signatures characterize various subsets of sample types. Importantly, we show that most of the AD-specific signatures are rare, each characterizing only 2 brain tissues or less. We demonstrate how biomarker-based diagnosis may overlook AD patients harboring distinct disease subtypes. Our results underscore the urgent need for unbiased, personalized AD diagnostics as well as personalized remedies in the future.

## 2. Methods

### 2.1. Study Design and Participants

This study utilized a gene expression dataset from 951 postmortem brain samples that were obtained from up to 17 brain regions of 107 subjects, deceased with varying AD-neuropathology severities. The dataset was published previously [12]. Up to 17 different brain regions were sampled in each patient: frontal pole (FP), occipital visual cortex (OVC), inferior temporal gyrus (ITG), middle temporal gyrus (MTG), superior temporal gyrus (STG), posterior cingulate cortex (PCC), anterior cingulate (AC), parahippocampal gyrus (PG), temporal pole (TP), precentral gyrus (PrG), inferior frontal gyrus (IFG), dorsolateral prefrontal cortex (DPC), superior parietal lobule (SPL), prefrontal cortex (PC), caudate nucleus (CN), hippocampus (Hi) and putamen (Pu). Each brain sample was profiled for over 44,000 transcripts using Affymetrix Human Genome U133A and U133B arrays (HG-U133A/B). Brain specimens that evidenced neuropathology other than that characteristic of AD were excluded [12]. 

### 2.2. Surprisal Analysis (SA) 

To characterize patient variability, we utilized surprisal analysis. SA decomposes the expression levels of the tested molecules into the expected expression levels at the steady state (i.e., the balanced, unconstrained state), and the deviations thereof due to environmental or genomic constraints [22,23]. Any genetic defect or epigenetic perturbation that prevents the cells from reaching the most stable state can be considered as a constraint. Each constraint significantly influences a subset of transcripts in a similar way by causing the collective deviations of the transcript levels (up or down) from their balanced levels. This group of co-varying transcripts is defined as an unbalanced process. To decompose gene expression levels into the levels at the steady (balanced) state and deviation thereof the following equation is utilized: $\ln X_{i}^{}(k)=\ln X_{i}^{o}(k)-\sum_{\alpha =1}G_{i\alpha }\lambda_{\alpha }(k)$ [23,24]. $X_{i}^{}\left(k\right)$ is the actual, experimentally measured expression level of gene *i* in sample *k*.
$X_{i}^{o}(k)$ is the expression levels at the steady, unconstrained state. In cases where $X_{i}^{}\left(k\right)\neq X_{i}^{0}$, we assume that the expression level of transcript *i* was altered due to constraints that operate on the system. The term $\sum_{\alpha =1}G_{i\alpha }\lambda_{\alpha }(k)$ represents the sum of deviations in expression level of transcript *i* due to the various constraints, or unbalanced processes that exist in the sample. 

The unbalanced processes, or constraints, are indexed by *α* = 1,2,3…. Several unbalanced processes may operate in each sample, and each transcript can participate in several unbalanced processes due to non-linearity of biological networks (contrary to clustering methods [15,25]). Singular value decomposition (SVD) is used as a mathematical tool to determine the two sets of parameters required in surprisal analysis to represent the unbalanced processes: (*1*) The *G*_i_*_α_* values (Appendix A), denoting the extent of the participation of each individual transcript *i* in the specific unbalanced process, *α*. Transcripts with significant *G*_i_*_α_* values (Appendix A) are considered to be affected by unbalanced process *α*. All transcripts with significant *G*_i_*_α_* values are grouped into the unbalanced processes group (Appendix A). Each unbalanced process is further interpreted using the David database as described below and shown in Appendix A. Note that the weight *G*_i_*_α_* is independent of *k*. Hence, the structure (transcript composition) of every process *α* remains constant. (*2*) The *λ_α_*(*k*) values, denoting the amplitude of each unbalanced process, in every sample *k*. The amplitude of an unbalanced process, *α*, determines whether process *α* is active in patient/sample *k*, and to what extent (see Appendix A).

In summary, the analysis uncovers the set of unbalanced processes that operate in the system, including the transcripts which are affected by these constraints (= unbalanced processes) and have thus deviated from their steady state levels. 

Complete details regarding the mathematical analysis have been described elsewhere [15,23]. 

### 2.3. Signs of $G_{i\alpha }$, and $\lambda_{\alpha }\left(k\right)$

The sign of $G_{i\alpha }$ (Appendix A) indicates the correlation or anti-correlation between transcripts in a particular process. For example, consider unbalanced process 1 in patient k, for which $\lambda_{1}\left(k\right)>0$. Transcripts 1, 2, and 3 were found to have different values, such as:$G_{\mathrm{transcript}1}=-0.01$,$G_{\mathrm{transcript}2}=0.01$, and $G_{\mathrm{transcript}3}=0$. This shows that in process 1 transcripts 1 and 2 are anti-correlated, i.e., deviate from their balanced levels in opposite directions, while transcript 3 is unaffected by process α. Note that each transcript can take part in several unbalanced processes.

The sign of $\lambda_{\alpha }\left(k\right)$ indicates the correlation or anti-correlation between the same processes in different patients/samples. For example, if process α is assigned the values in samples 1, 2, and 3: $\lambda_{\alpha }(\mathrm{sample}\;1)=28$, $\lambda_{\alpha }(\mathrm{sample}\;2)=0$, and $\lambda_{\alpha }(\mathrm{sample}\;3)=39$, it means that this process is active in sample 1 and 3 in the same direction (i.e., the transcripts affected by this process deviate to the same directions in samples 1 and 3), while it is not active in sample 2. 

To find the actual change in expression level for each transcript *i* in every sample *k* we calculate the product $G_{i\alpha }\lambda_{\alpha }\left(k\right)$, which can be positive (indicating upregulation) or negative (indicating downregulation). Note that the product $G_{i\alpha }\lambda_{\alpha }\left(k\right)$ only denotes the change in expression level of the transcript *i* that occurred due to unbalanced process *α*. To calculate the complete change in expression level, the sum of contributions of different unbalanced processes, α=1,2,3…, is calculated: $\sum_{\alpha }G_{i\alpha }\lambda_{\alpha }\left(k\right)$.

### 2.4. The Biological Meaning of Each Unbalanced Process

To assign a biological meaning to each unbalanced process, transcripts were categorized using David software according to the Gene Ontology (GO) database. Some transcripts were involved in one unbalanced process, whereas others participated in 2 or more unbalanced processes. Each unbalanced process can include multiple (sometime overlapping) biological categories (Appendix A).

### 2.5. Determination of the Number of Significant Unbalanced Processes 

The number of significant constraints was determined as described previously [15,21]. Briefly, the analysis of the 951 samples provided a 951 × 951 matrix of $\lambda_{\alpha }(k)$ values [23], such that every column in the matrix contained 951 values of $\lambda_{\alpha }(k)$ for 951 samples, and corresponded to an unbalanced process (Appendix A). However, not all unbalanced processes are significant. Our goal was to determine how many unbalanced processes were needed in order to reconstruct the experimental data, i.e., for which value of *n*: $\ln X_{i}^{}(k)\approx -\sum_{\alpha =0}^{n}G_{i\alpha }\lambda_{\alpha }\left(k\right)$. To find *n*, we performed the following two steps:

(1) Processes with significant amplitudes were selected: To calculate threshold limits for $\lambda_{\alpha }\left(k\right)$ values (presented in Appendix A) standard deviations in gene expression levels of the 100 most stable transcripts in this dataset were calculated (e.g., those with the smallest standard deviations values). Those fluctuations were considered as baseline fluctuations in the population of the patients which were not influenced by the unbalanced processes. Using standard deviation values of these transcripts the threshold limits were calculated as described previously [26]. The analysis revealed that from α=8 the $\lambda_{\alpha }\left(k\right)$ values become insignificant (i.e., do not exceed the noise threshold), suggesting that 7 unbalanced processes are enough to describe the dataset. 

(2) Reproduction of the experimental data by the unbalanced processes was verified: To verify that the number of processes identified in step 1 is correct, we plotted $\sum_{\alpha =1}^{n}G_{i\alpha }\lambda_{\alpha }(k)$ against $\ln X_{i}^{}(k)$ for different transcripts and for different values of *n*, and examined the correlation between them as *n* was increased. An unbalanced process, α=n, was considered significant if it improved the correlation significantly relative to α=n−1. Figure 4B shows that increasing *n* from 7 to 10 did not significantly affect the correlation between the theoretical and experimental data for different samples.

### 2.6. Calculation of Barcodes

The barcodes presented in Appendix A were generated as described previously [17]. Briefly, barcodes represent a sample-specific combination of active unbalanced processes. Barcodes were generated using a custom python script. For each sample, $\lambda_{\alpha }\left(k\right)$ values (α=1,2,3,…,7) were normalized as follows: If $\lambda_{\alpha }\left(k\right)>24$ (and is therefore significant according to calculation of threshold values), then it was normalized to equal 1; if $\lambda_{\alpha }\left(k\right)<-24$ (significant according to threshold values as well), then it was normalized to equal −1; and if $-24<\lambda_{\alpha }\left(k\right)<24$, then it was normalized to equal 0. Appendix A lists 52 unique barcodes that were calculated and found to repeat themselves in the 951 brain samples. The results are shown graphically in Figures 5 and 6. 

## 3. Results

### 3.1. A Large-scale Dataset was Selected for Study

We investigated a large-scale gene expression dataset containing 951 postmortem brain samples that were obtained from up to 17 brain regions of 107 subjects, either with varying severities of AD neuropathology or non-demented [12]. 

All subjects ranged from 60 to 100 years of age. Of the 107 subjects, 22 were characterized as non-demented. The remaining 85 AD-bearing subjects were divided into three categories according to their pathological severity stage: possible, probable, or definite [12]. 

The total of 951 samples included 737 AD samples and 214 control, non-demented samples. Each sample represented a specific brain region of a specific patient. 

The complete list of samples, including sample-specific information such as subject ID, age, pathological stage, and brain region, can be found in Appendix A. 

### 3.2. An Overview of Our Approach

We wished to explore the variability among the AD brain samples in the dataset, aiming to shed light on AD patient characterization. The question we sought to address is whether different AD patients can be identified based on specific molecular characteristics that repeat themselves throughout the population.

The search for new biomarkers was defined as one of the central future goals to advance accurate AD diagnostics and treatment [27,28,29]. Research/diagnostics routine in AD includes a search for novel AD biomarkers (mutations, variants (e.g., ApoE)) or testing expression levels of known AD biomarkers (e.g., amyloid-β (Aβ), Tau, ApoE, Bin1, etc.) (Figure 1A,B). These are searched for on genomic, transcriptomic, or proteomic levels. Next, patient diagnosis and classification is performed based on upregulated expression levels or mutations of the biomarkers discovered (Figure 1C). 

Based on the high phenotypic variability recognized in AD (see Introduction), we postulated that a finite set of AD-specific altered biomarkers may not suffice to define AD pathology. Therefore, rather than selecting a few biomarkers for examination (Figure 1A), our approach employed high-throughput profiling for every patient. Next, surprisal analysis (SA) was utilized to decipher the altered transcriptional signatures in the patient population (Figure 1D; see detailed explanation in the Methods section and references [15,23]). The analysis assumes that AD tissues are biological systems in which the balanced homeostatic state has been disturbed due to genomic and/or environmental factors, or constraints. Different combinations of genomic and/or environmental constraints can operate in different brains, giving rise to variability in gene expression patterns within the patient population. Every constraint can alter a part of the gene network structure in the AD brain, such that a specific group of transcripts undergoes coordinated changes in expression levels, generating an unbalanced process. In other words, an unbalanced process is the subnetwork that was altered due to the constraint. SA discovers the unbalanced processes that repeat themselves in the entire population of patients (Figure 1D) and then determines which of these processes have emerged in each and every AD sample (Figure 1E). Figure 1 emphasizes the importance of deciphering the accurate transcriptional signature. For example, consider biomarker B (Figure 1). In the traditional routine, mutation/overexpression of this biomarker would be measured in order to diagnose AD patients. Patients 1 and 2 both overexpress this biomarker and would thus be classified as similar (Figure 1B,C). SA, however, shows a broader picture: patient 1 harbors 3 distinct unbalanced processes, highlighted in black, green, and yellow (Figure 1E), while patient 2 harbors only the black unbalanced process (Figure 1E). In patient 1 the upregulation of biomarker B is associated with 2 distinct unbalanced processes (black and green, Figure 1E), while in patient 2 the upregulation of biomarker B is attributed only to the black process (Figure 1E). Thus, despite similar expression levels of the biomarker, the AD tissues in these patients differ and may therefore demand different diagnoses and different modalities of treatment (Figure 1F). 

We suggest that in order to expand our understanding of the disease, the complete set of unbalanced networks should be determined for every tissue. Deciphering the complete altered transcriptional network in each AD sample/patient can be especially beneficial for personalized drug design in the future, in which central proteins from distinct unbalanced networks can be targeted to reduce the altered signaling flux (e.g., as recently suggested by us for the treatment of cancer [17]).

### 3.3. 951 AD-related and Normal Brain Samples Harbor A Similar Balanced State Process

When we examined the reference (balanced) state and the transcripts that were found in this state, we found that the reference state remained essentially the same across the AD and non-demented samples (Figure 2), as indicated by the similar amplitudes across all 951 samples ($\lambda_{0}\left(k\right)$ represents the amplitude (= importance) of the steady state, *α* = 0, in each sample *k*; see Methods). 

A subset of the transcripts comprising the steady state (i.e., transcripts with the highest steady state weights, *G_i0_*; see Methods, Appendix A) were the most stable transcripts, as they did not change their expression levels across different samples, and consequently, did not participate in any of the unbalanced processes, denoted *α*=1, 2…. 

The steady state’s most stable transcripts were categorized to the basic homeostatic functions of the cell such as protein translation, RNA synthesis, and ATP synthesis (Appendix A; tab “steady state G0”). 

Additional categories enriched in the steady state of the current dataset were related to axonogenesis and neuron development, providing further characterization of the homeostatic functions of the elderly AD and non-demented brains (Appendix A).

### 3.4. Similar Gene Expression Levels may Overlook Differences between Individual Samples and Patients

We searched the dataset for the expression levels of a few known AD-associated gene biomarkers: APOE (apolipoprotein E), BIN1 (Myc box-dependent-interacting protein 1), PTK2B (protein tyrosine kinase 2 beta) and PLD3 (phospholipase D3). It has been shown that APOE and BIN1 are usually upregulated, while PTK2B and PLD3 are downregulated in AD patients [28,30,31]. 

APOE is considered a major genetic risk factor for Alzheimer’s disease [32,33]. This biomarker is usually tested at either genetic (e.g., gene variants of APOE [7]) or gene expression levels [11,30,33]. Although different APOE haplotypes may influence the expression patterns of APOE gene in a different manner, it is usually the increased protein activity and/or induced expression of the APOE gene which are linked to the increased risk to develop AD [11,33]. Thus, despite unavailability of the information regarding the APOE gene variants in this dataset, patient-specific expression levels of APOE may provide interesting insights.

Figure 3 presents the expression levels of these four AD biomarkers in the 951 samples tested. 

All four biomarkers demonstrated varying degrees of expression in the different samples, as expected. It is important, however, to note that only 10% and 2% of the definite AD brain samples demonstrated significant upregulation of APOE and BIN1, respectively (Figure 3; the number and percentage of definite AD samples in the gray area is denoted). Additionally, only 6% and 12% of the definite AD brain samples demonstrated significant downregulation of PLD3 and PTK2B, respectively (Figure 3). Hence, most of the brain samples did not “behave” as expected in terms of these four biomarkers, suggesting that biomarker-based identification of AD patients may lack important patient-specific information, a notion that may have significant implications on future personalized AD diagnosis and therapy.

### 3.5. The 951 AD and Normal Brain Samples can be Characterized by Seven Unbalanced Processes

We deciphered the complete set of unbalanced processes that emerged in the dataset and in each individual sample.

Using SA and error analysis (described in the Methods and [15]), we found that seven unbalanced processes repeated themselves across the 951 AD and non-demented samples (Figure 4A shows zoom-in images on selected transcripts in some of the unbalanced processes; the full data regarding the unbalanced processes can be found in Appendix A). These seven processes sufficed to reproduce the experimental gene expression data obtained from those samples (Methods and Figure 4B), signifying that SA analysis achieved considerable compaction of the data. 

Each sample in the dataset was characterized by a small subset of these seven unbalanced processes, typically 1-3 processes. We inspected, for example, three samples obtained from three different AD patients: sample 919 (GSM2234520), which was obtained from the Pu of subject 1006 (diagnosed possible AD); sample 944 (GSM2234550), which was obtained from Pu of subject 66 (diagnosed definite AD); and sample 141 (GSM2233767), obtained from the ITG of subject 616 (diagnosed possible AD). In all three samples, APOE and BIN1 biomarkers were upregulated relative to their median expression levels (Figure 4C). However, SA revealed that the samples were biologically different: sample 919 was characterized by a combination of processes 1, 3, and 6; sample 944 was characterized by processes 1 and 6; and sample 141 harbored only process 5 (Figure 4D). In samples 919 and 944, the upregulation of APOE was associated with process 1, while in sample 141 it was associated with process 5 (Figure 4A,D). The upregulation of BIN1 was associated with process 6 in samples 919 and 944, whereas in sample 141 the upregulation of BIN1 was associated with process 5 (Figure 4A,D). 

PTK2B and PLD3 were both downregulated relative to their median expression levels in samples 919 and 944 (Figure 4C). However, PLD3 downregulation was associated with processes 1 and 3 in sample 919, and with only one process—process 1—in sample 944 (Figure 4A,D). 

Hence, we show that samples from the same/different brain regions may seem very similar in terms of their transcriptional expression, but nevertheless still differ from one another. Therefore, a comprehensive examination of the complete altered transcriptional signature is required.

In general, all four selected biomarkers were each found to participate in a few distinct unbalanced processes: APOE was found to participate in unbalanced processes 1, 2, 4, and 5; BIN1 in unbalanced processes 4, 5, and 6; PLD3 in unbalanced processes 1,2, 3, 5, 6, and 7; and PTK2B in unbalanced processes 1, 5, and 7 (Appendix A, Figure 4A exemplifies this point by showing the simultaneous participation of those transcripts in several processes harbored by the samples 141, 919, and 944). Hence, measuring the expression levels of a few transcripts in each patient may not suffice, as the deviations in expression levels of the different biomarkers can arise due to different active unbalanced processes in every sample. AD patients can have similar expression levels of AD biomarkers, while harboring different transcriptional signatures (i.e., different sets of active unbalanced processes, as exemplified above), and thus different AD molecular phenotypes. Therefore, an in-depth evaluation of the sample-specific altered transcriptional network is essential in order to adequately comprehend the molecular variability among AD patients and samples.

### 3.6. Validation of the Robustness of the Analysis

We validated the robustness of our findings by several means. 

First, we examined whether the seven unbalanced processes that were identified in the 951 samples were relevant in a smaller subset in which 50 random patients out of 107, constituting together 451 samples, were selected. The weights of the transcripts ($G_{i\alpha }$) and the amplitudes of the unbalanced processes ($\lambda_{\alpha }(k)$) identified in the small dataset were compared to those identified in the full dataset. For each unbalanced process *α*, the weights of the transcripts from the smaller dataset were highly correlated with the weights of the transcripts from the full dataset (Appendix A). Hence, the unbalanced processes that were found in the small dataset match those found in the full dataset. The amplitudes of the unbalanced processes ($\lambda_{\alpha }(k)$) were found to highly correlate as well (Appendix A). 

To validate that the dataset of 951 was not over fitted we divided the original dataset into two halves and analyzed each subset separately. The analysis yielded the same results: the same unbalanced processes appeared in two separate datasets when those subsets were analyzed independently and then compared to either the original dataset (Appendix A) or one to another (Appendix A). 

Next, we asked whether we could identify the same unbalanced processes and the patient-specific amplitudes by analyzing only a subset (~half) of the transcripts. We analyzed 22,351 transcripts (instead of ~44,000 transcripts) in 951 samples and found that those transcripts were assigned to the same unbalanced processes. The amplitudes and the weights of the transcripts which were identified in the original dataset highly correlated with the amplitudes and the weights of the transcripts which were identified in the smaller dataset showing that a smaller number of genes can characterize the AD dataset in a similar manner (Appendix A)**.** This result can be explained by the existence of regulatory mechanisms in the cells that limit the change in the expression levels of molecules, i.e., molecules in the cells are not free to vary independently, but rather are dependent on the expression levels of other molecules. It is for this reason that the addition/subtraction of the input transcripts in the analysis does not significantly change the weights of the transcripts and thus the nature and the composition of the unbalanced processes.

The fact that independent analyses of sub-datasets (containing either part of the samples or part of the transcripts) yield essentially the same results as analysis of the full dataset, suggests that the dataset selected for study is large enough to yield creditable information regarding the altered transcriptional signatures in the AD samples tested. 

An additional issue for validation was the number of normal samples vs. the number of AD samples. The dataset tested contained 22 non-demented subjects (214 samples) and 85 AD subjects (737 samples). To verify that the difference in the number of samples did not affect our results, we performed a separate analysis of only the AD samples. We show in Appendix A that the same unbalanced processes were identified in this analysis, suggesting that the normal samples and their amount do not influence the conclusions of the analysis.

### 3.7. The Unbalanced Processes Identified by SA Capture Known AD-related Biological Characteristics 

To verify whether the division of the transcripts into unbalanced processes by SA corroborated with previous knowledge regarding the biological activity of the transcripts in AD, we utilized the David database [34], as described above for the steady state process, to assign a biological meaning to each unbalanced process (Appendix A). 

Unbalanced process 1, the most dominant process in the dataset, appearing in 170 samples out of 951, was found to be especially dominant in CN, Hi, and Pu brain regions (Appendix A; unbalanced process 1 was assigned a negative amplitude in CN, Hi, and Pu. See the Methods section for an explanation on how the signs of G_iα_ and $\lambda_{\alpha }\left(k\right)$ should be interpreted). Transcripts affected by unbalanced process 1 were found to participate in multiple enriched biological categories, including downregulation of memory, the ability to learn, and in proliferative functions of the cell (Appendix A). Certain other regions of AD-related brains and non-demented tissues were found to harbor this process as well (Appendix A). Similar percentages of AD and non-demented tissues were found to harbor this process (Figure 5A). Additionally, the majority (>70%) of the non-demented tissue samples harboring process 1, were obtained from people who were over 80 years old, suggesting that this process is generally related to aging. 

Another biological feature associated with unbalanced process 1 was dysregulation of intracellular calcium signaling, mainly in Hi, Pu, and CN brain regions (Appendix A; tab “process 1 G1 > 0.01”). Calcium modulates many neural processes including synaptic plasticity and apoptosis. Disruption of calcium regulation in the endoplasmic reticulum mediates the most significant signal transduction cascades that are associated with aging and AD [35,36]. Process 1 was found to be associated with downregulation of transcripts involved in calcium ion transport, calcium ion-regulated exocytosis of neurotransmitter and positive regulation of calcium ion-dependent exocytosis. Moreover, signaling pathways such as negative regulation of apoptosis and the mitogen-activated protein kinase (MAPK) pathway which are known to be up-regulated in cancer and to play a central anti-apoptotic role in tumor progression, were found to be downregulated in process 1 in Hi, Pu, and CN brain regions (Appendix A; tab “process 1 G1 > 0.01”), in accordance with their reported role in learning and memory formation [37].

Brain regions with positive $\lambda_{1}\left(k\right)$ values (Appendix A, Appendix A; 145 samples originating from 42 subjects), were characterized by another group of downregulated transcripts which were similarly involved in signal transduction, positive regulation of apoptotic cell clearance, and nervous system development (Appendix A; tab “process 1 G1 < −0.01”). This result demonstrates that although the transcripts associated with unbalanced process 1 deviate to opposite directions in patients with positive $\lambda_{1}\left(k\right)$ values vs. those with negative $\lambda_{1}\left(k\right)$ values (upregulated transcripts in the patients with positive $\lambda_{1}\left(k\right)$ values are downregulated in the patients with negative $\lambda_{1}\left(k\right)$ values, and vice versa), in both cases they eventually represent a similar AD/aging phenotype characterized by induced apoptosis, reduced proliferative signaling, and attenuated neurological processes. 

Process 2 was found to be dominant in a group of 135 AD and non-demented brain samples (with negative $\lambda_{2}\left(k\right)$ values; originating from 47 subjects). These samples originated from multiple brain regions including FP, OVG, ITG, and more (Appendix A). This process included (but was not limited to) downregulated transcripts involved in small GTPase mediated signal transduction, intracellular signal transduction, synaptic transmission, learning, and phosphorylation (Appendix A; tab “process 2 G2 > 0.01”). 

The biological interpretation of the other unbalanced processes (indexed 3–7) can be found in Appendix A.

### 3.8. Exploring Sample-Specific Transcriptional Signatures 

We asked whether specific unbalanced processes could distinguish between AD and non-demented tissues and therefore constitute extended AD-associated biomarkers. Should such unbalanced processes exist, i.e., such that appear in the AD tissues and not in the non-demented tissues, or vice versa, these processes may be utilized to identify AD patients once a technology to obtain brain biopsies, or non-invasive brain diagnostic techniques, such as liquid biopsies [38], will be developed.

Interestingly, the three most dominant unbalanced processes (indexed 1, 2, and 3; Appendix A) could not distinguish between AD and non-demented brain samples, as they appeared in similar percentages of normal and demented samples (Figure 5A). The remaining unbalanced processes, indexed 4–7, each appeared in a small number of samples (less than 3% of the samples) and were therefore not considered as distinguishing processes (Figure 5A). 

As mentioned above, every individual sample harbored a specific set of ~1-3 active unbalanced processes, namely an altered transcriptional network signature (Appendix A). We hypothesized that the complete 951 sample-specific transcriptional signatures may provide the ability to identify demented vs. non-demented samples.

Using SA parameters, namely the amplitudes of the unbalanced processes that were calculated for each brain sample ($\lambda_{\alpha }\left(k\right)$; Appendix A), brain sample-specific transcriptional signatures for each AD and non-demented brain sample, *k,* were identified [15] (see Methods). 

To simplify the representation of the sample-specific sets of unbalanced processes, we computationally transformed the sample-specific combinations of unbalanced processes into personalized schematic barcodes (Figure 5B, Appendix A, and Methods). The sample-specific barcodes of samples 919, 944, and 141 (discussed above and shown in Figure 4D) are shown in Figure 5B.

In total, we found 52 unique subsets of 1-3 unbalanced processes (out of seven) that repeated themselves across the 951 samples (737 diseased samples and 214 non-demented samples). The null barcode, indexed #1, represented samples which did not include any active unbalanced processes, but rather only the steady state process (Appendix A, tab “list of 52 barcodes”). 

Thirty of the barcodes characterized only AD samples, but not normal, non-demented samples (Appendix A). Importantly, rare altered network structures that appeared in a very small number of samples were not overlooked. We found that 21 of the 30 barcodes each characterized only 2 AD brain samples or less (we define these barcodes as rare; Appendix A). Interestingly, we noted a high correlation between the rarity of the barcode and its specificity to AD samples (i.e., the majority of the rare barcodes are AD-specific; Appendix A), emphasizing that AD samples are highly heterogeneous in terms of their altered transcriptional networks, each sample being highly unique.

### 3.9. Different Brain Regions in the Same Patient may Harbor Distinct Transcriptional Signatures

Interestingly, we found that the brain of each AD patient can harbor several barcodes, each representing a different region in the brain, unraveling an additional layer of complexity existing in AD disease (Figure 6A, an example for subject 111 is shown; the complete information for all subjects in the dataset is presented in Appendix A, tab “Sample-specific barcodes”, summarizing the barcodes’ appearances in each brain region and pathological condition). Thus, similar to cancer, AD pathology can be characterized as an intra-brain heterogeneous disease. 

Although brains of many AD patients harbored multiple barcodes per brain, most of those barcodes (in ~60% patients) were assembled from the same 2–3 unbalanced processes that repeated themselves in different barcodes, i.e., different combinations. For example, the brain of AD patient 786 was characterized by six different barcodes (Appendix A). However, those barcodes were assembled from only three unbalanced processes (processes 1, 2, and 3). The brain of patient 869 was characterized by five distinct barcodes, which were assembled from different combinations of processes 1, 2, and 7 (See Appendix A for more examples). 

These results suggest that highly variable inter- and intra-heterogeneous gene expression datasets can eventually be compacted to a few unbalanced processes per patient. Reorganization of the highly heterogeneous datasets into simple combinations of ~2–3 diseased processes per sample can simplify the diagnostics of AD and guide personalized treatment in the future [17]. We hypothesize that targeting several proteins involved in distinct unbalanced processes [17] can serve as an effective personalized AD treatment. 

Figure 6B demonstrates the intra-region heterogeneity in each brain region. We analyzed the significance of each barcode in each brain region by calculating the ratio between AD vs. non-demented samples that harbored the specific barcode in that brain region. To achieve significant results, only the barcodes characterizing at least 10% of the diseased or non-demented samples were included in the computation (Appendix A). Using this criterion, it became evident, for example, that barcode 2 was enriched mostly in Hi, CN, PG, PCC, and Pu of AD patients (Appendix A). This barcode included only process 1 (Appendix A) and was characterized by reduced expression of the transcripts involved in brain functions and cancer growth (Appendix A, tab “Process 1 G1 > 0.01”) and induced expression of transcripts involved, for example, in immune response and negative regulation of protein serine/threonine kinase activity (Appendix A, tab “Process 1 G1 < −0.01”). 

Importantly, however, additional barcodes may appear in the same brain region (e.g., barcode 26 in CN (Appendix A). 

Moreover, the same barcode can be found in other brain regions of non-demented tissues. For example, barcode 2 characterized mostly non-demented tissues in AC (Appendix A). 

We also found that certain barcodes (see for example #10, 50, and 52 in Appendix A) characterized both AD and non-demented samples. 

While there are a few brain regions known to be more relevant to AD pathology [12,39,40], recent studies have shown that additional brain regions undergo changes as well, in some cases resulting from non-pathological effects of the cells [39,40]. SA analysis reveals that transcriptional alterations can occur in all 17 brain regions tested (Appendix A; tab “sample-specific barcodes”). The nature and role of these alterations merit further investigation, which is out of the scope of this study.

### 3.10. A Spectrum of Distinct Barcodes Characterizes the Brain Samples in the Dataset

We mapped the different barcodes identified by SA based on their frequency of expression in the different sample types—non-demented, possible AD, probable AD, and definite AD (Figure 7). 

Some “blocks” of barcodes representing solely one type of sample were visible, e.g., barcode 42 characterizes only normal samples (Figure 7A), barcodes 6, 16, and 18 characterize only probable AD samples (Figure 7C), and barcodes 9, 11, 37, 43, and 45 only definite AD (Figure 7D). However, the map is complex, because it shows that a spectrum of barcodes, each characterizing a certain subset of the sample types, was identified by SA. For example, barcode 4 characterizes non-demented samples, possible, and probable AD equally, while not characterizing any definite AD samples (Figure 7). Barcodes 32, 39, and 47 characterize possible and definite AD samples with an equal frequency (Figure 7). Barcodes 23, 34, and 36 characterize possible and probable AD samples (Figure 7). Various barcodes each characterize all types of samples, e.g., 46, 30, 25, 10, and more (Figure 7).

To conclude, our attempts to find commonalities among AD patients were unsuccessful, underscoring an important conclusion: AD appears to be a highly variable disease, characterized by multiple molecular aberrations. Our data demonstrate that the molecular picture in AD is of high complexity. This may also be partly since these samples were obtained from the brains of elderly subjects, and therefore harbor transcriptional alterations related to the aging process.

We suggest that our conception of AD pathology should shift from biomarker-based characterization to multi-modal characterization, possibly integrating biochemical and behavioral examinations. Once a method to accurately diagnose AD patients is developed, it will be vital to replace the search for commonalities among AD patients, with the identification of the differences between them, in order to eventually tailor the treatment to the specific patient.

## 4. Discussion

Various clinical fields such as oncology [41], cardiology [42], and neurology [43,44] have refocused their efforts from a traditional “one-size-fit-all approach” to explore more personalized-based approaches. Accordingly, the classification of disease subtypes is currently a key challenge. One approach to define such subtypes is based on the translation of “omic” profiles from multiple patients into useful personalized information that will allow accurate patient classification and support a future transition to precision medicine. 

In this study we implemented an unbiased information-theoretic approach to explore the heterogeneity among a population of AD patients. The approach we presented herein allows characterizing the altered transcriptional network structure of each patient individually. We have recently demonstrated, using cancer models, the validity of the network signatures identified by SA by designing potent patient-specific targeted drug combinations based on these networks [17,21]. We have shown that in several cancer types the SA-based therapy induced higher rates of cell death than the clinically prescribed therapy [17]. 

Here we utilized SA to study 951 AD and non-demented samples. We found that all samples, AD as well as non-demented, shared an invariant steady state process (Figure 2). A robust, invariant steady state was previously found in cancer vs. normal samples in several studies [15,22,45]. In our recent studies, the steady state transcripts of normal and cancer tissues were assigned to biological categories similar to those described herein [22,45]. The finding that AD and non-demented tissues share a similar balanced state is significant, because it suggests that in order to characterize AD heterogeneity, only unbalanced processes should be examined, while transcripts that participate in the steady state can be disregarded. This notion reduces the scope of transcripts that should be tested and may simplify AD research, diagnosis, and eventually treatment.

AD genotypes (such as the combinations of APOE variants) are commonly tested and linked to the risk for Alzheimer’s disease. However, it is the gene products (transcripts and proteins) that are eventually responsible for the manifestation of AD phenotype. Thus, relating gene expression levels of known AD biomarkers to molecular variability of AD patients can provide important insights regarding patient classification. Here we demonstrated that similar transcript expression levels of AD biomarkers can emanate from different sets of unbalanced processes. This observation suggests that in order to achieve comprehensive AD molecular characterization, relationships of a certain biomarker with other measured transcripts (obtained through calculation of unbalanced processes) should be examined. 

We found that the heterogeneous collection of samples could be compacted and described by seven transcriptional unbalanced processes. Each brain sample was characterized by a sample-specific set of 0-3 unbalanced processes out of seven. This sample-specific combination provides an individualized transcriptional signature (= barcode) for every tested sample. 

The finding that ~40% of the samples were not characterized by any of the unbalanced processes is interesting, because it suggests that transcriptomic evaluation of AD samples may not suffice to uncover the molecular processes gone awry in all cases of AD. In the future, when large datasets consisting of multiple samples, profiled simultaneously by transcriptomics, proteomics, and genomics, will be available, it will be important to integrate the transcriptional signatures with the proteomic signatures in AD, potentially allowing higher accuracy in the diagnosis, more precise categorization, and personalized treatment of AD patients.

Analysis of intra-brain heterogeneity revealed that AD brains can harbor different barcodes in distinct brain regions, resembling intra-tumor heterogeneity in cancer [46]. However, although different AD signatures were found in distinct brain regions of a certain patient, in the majority of the cases those signatures were assembled from different combinations of the same few unbalanced processes (3 or less). This suggests that not only a sample from a certain brain region, but also the entire brain can be characterized by a few unbalanced processes which may provide guidance on how personalized treatment should be designed.

In total, we found that 52 unique signatures recurred across 951 different AD/non-demented samples. Thirty of them were unique to diseased samples (Appendix A). Our findings show that no single barcode can be used to diagnose all cases of AD. Rather, the barcodes that we found suggest that AD can be viewed as a collection of different AD subtypes, each molecularly distinct. 

We show herein that many of the AD samples are rare in terms of the altered transcriptional network, or barcode, that they harbor, i.e., many transcriptional signatures are shared by only two samples or less. Additionally, although certain diseased barcodes appeared with higher probability in certain brain regions (e.g., barcode 2 in CN, PCC, and PG; Appendix A), many other barcodes could appear in the same brain regions of various patients (Figure 6). 

This points to an important message we wish to convey—when inspecting AD patients and attempting to treat them in an individualized manner, instead of searching for the commonalities among the patients (e.g., in the form of biomarkers), it may be more effective to search for the differences among them. Knowledge of what differentiates a specific subgroup of patients from other subgroups may grant us the knowledge of how to correctly tailor the treatment to the patients that suffer from this type of the disease.

Accumulating data have shown that certain brain regions play important roles in AD pathology, and that the progression of the disease to different regions in the brain may characterize different stages of the disease [12,39,40]. However, our focus in this study was to identify the sample-specific altered transcriptional signature at the time of sampling. This is because our interest was in highlighting the molecular differences between the samples, which may merit personalized treatment regimes. The evolutionary aspect was out of the scope of our study and is an issue for further research.

The dataset studied herein contained only sporadic cases of AD. In the future, when large datasets of familial AD samples will be available, it may be insightful to perform a comparative study between sporadic and familial AD. For example, it would be interesting to check whether the distribution of barcodes remains similar or not, or whether there are transcriptional signatures specific to familial cases and vice versa. The comparison between sporadic and familial cases of AD is particularly important as mouse AD models were engineered to express fAD-causing mutated human genes. Such comparison will provide insights into the relevance of these models to the study of sporadic cases which constitute the vast majority of AD cases. 

To conclude, we suggest that our conception of AD pathology should be shifted, to view AD as a disease that can be characterized by various distinct molecular alterations. We believe that a strategy which combines advanced computational and biochemical characterization as well as behavioral examination will pave the way for the development of novel patient-specific diagnoses and remedies for one of the most prevalent and devastating maladies of the 21st century. 

## 5. Conclusions

In this study we present an information-theoretic approach that allows translating large-scale heterogeneous molecular-level information from 951 non-demented and AD-bearing tissues into seven altered transcription sub-networks, named unbalanced processes. Those seven unbalanced processes constructed 52 distinct transcriptional signatures, which were repetitive across the 951 brain tissues. We found that many of the AD-specific transcriptional signatures were rare, and appeared only in two brain samples or less. We show that biomarker-based diagnostics may classify molecularly distinct AD tissues as similar. Thereby, we propose that the AD research community should adopt a new viewpoint on AD pathology, recognizing the high complexity and molecular variability of the disease, and understanding that it is the differences between the patients that we should be searching for, rather than the commonalities among them.

## Figures and Tables

**Figure 1 biomolecules-10-00503-f001:**
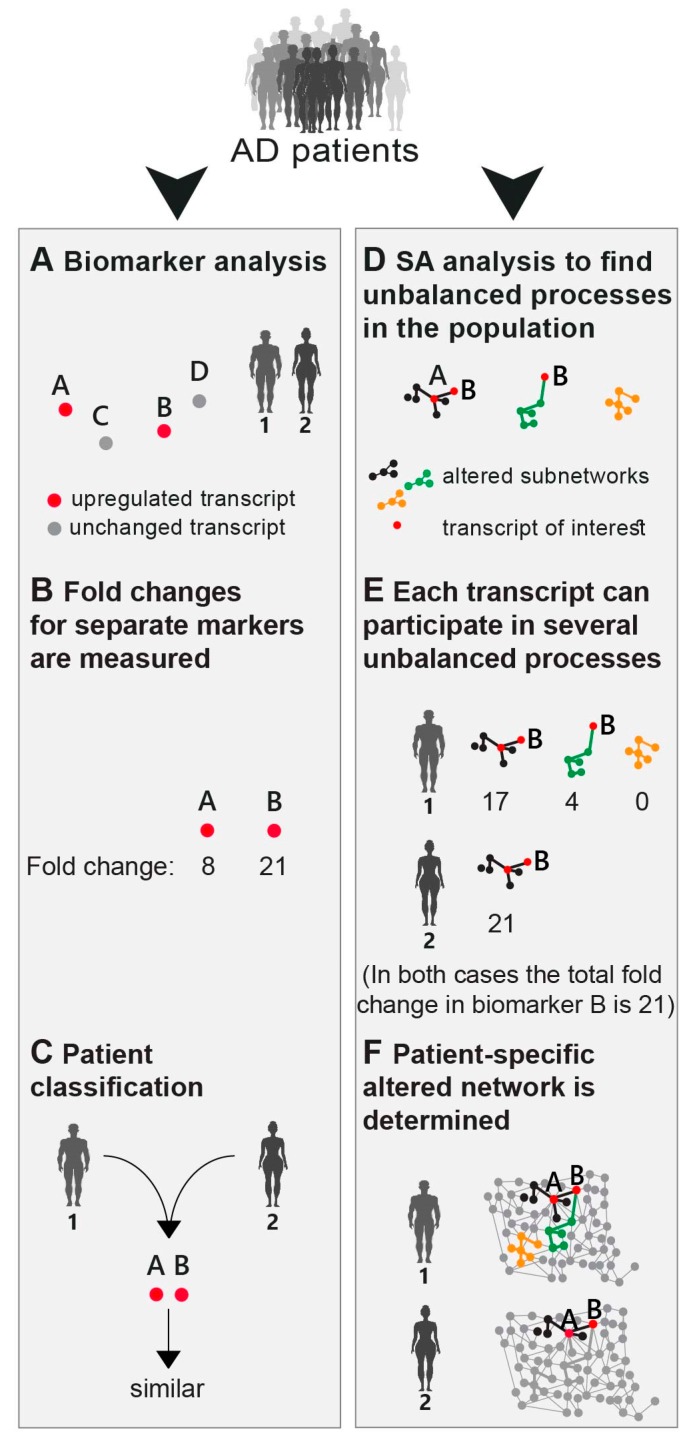
An overview of the approach. (**A**) The common strategy today is to look for novel biomarkers that would classify Alzheimer’s disease (AD) patients in a more accurate, patient-specific manner. In the illustrated example, four AD biomarkers were tested (genes *A–D*). Based on the expression levels of those markers (**B**) the patients are classified. Patients 1 and 2 in this example have similar expression levels of biomarkers A and B, and therefore would be classified as molecularly similar (**C**). We explore AD pathology in an unbiased manner (**D**–**F**). The workflow of our approach consists of patient-specific “omic” profiling, followed by surprisal analysis (**D**), aiming to decipher not only altered transcripts/proteins, but also the structure of the altered network, namely the patient-specific altered transcriptional signature (**E**,**F**). This signature is composed of distinct unbalanced processes, each resulting from a constraint that operates on the system (see main text). Overexpression of the biomarker B in patient 1 is associated with the black and green processes, whereas in patient 2 it is upregulated due to only the black process (**D**–**F**).

**Figure 2 biomolecules-10-00503-f002:**
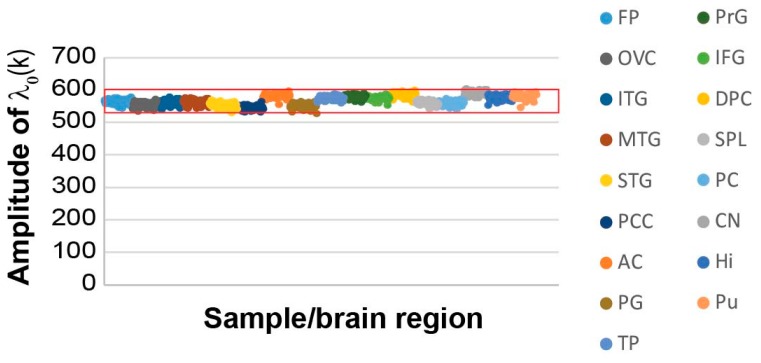
The studied 951 non-demented and AD-bearing brain samples harbor a similar steady state process. The amplitudes, λ_0_(k), of the steady state for all samples are presented, demonstrating an invariant amplitude across the 951 samples. The data is color-coded to show the different brain regions (frontal pole (FP), occipital visual cortex (OVC), inferior temporal gyrus (ITG), middle temporal gyrus (MTG), superior temporal gyrus (STG), posterior cingulate cortex (PCC), anterior cingulate (AC), parahippocampal gyrus (PG), temporal pole (TP), precentral gyrus (PrG), inferior frontal gyrus (IFG), dorsolateral prefrontal cortex (DPC), superior parietal lobule (SPL), prefrontal cortex (PC), caudate nucleus (CN), hippocampus (Hi) and putamen (Pu)). The red box marks the error limits, which were calculated for the steady state based on the expression values of the transcripts that were found to participate in the steady state (Appendix A) [26].

**Figure 3 biomolecules-10-00503-f003:**
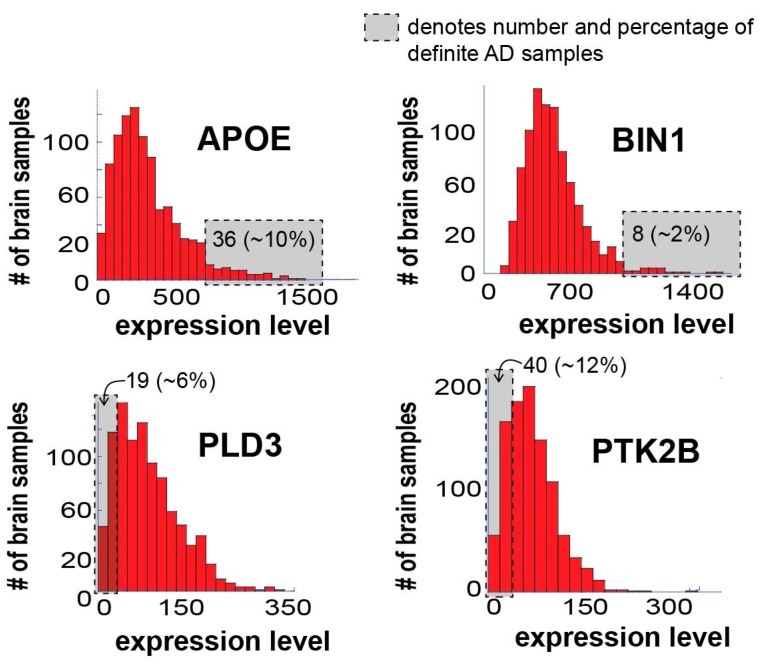
Expression levels of known AD biomarkers vary significantly in different AD patients and most samples do not “behave” as expected in terms of these biomarkers. Four known AD biomarkers were selected for the demonstration of the concept: APOE, BIN1, PLD3, and PTK2B. The figure shows a histogram of expression levels for each of the biomarkers. In each graph, the gray box denotes the number of definite AD samples in the boxed area, including their percentage out of the total definite AD patients in the dataset. Only 36 definite AD samples (10%), significantly overexpress APOE; only 8 definite AD samples (2%) significantly overexpress BIN1; only 19 definite AD samples (6%) demonstrate significant downregulation of PLD3; and only 40 definite AD samples (12%) demonstrate significant downregulation of PTK2B.

**Figure 4 biomolecules-10-00503-f004:**
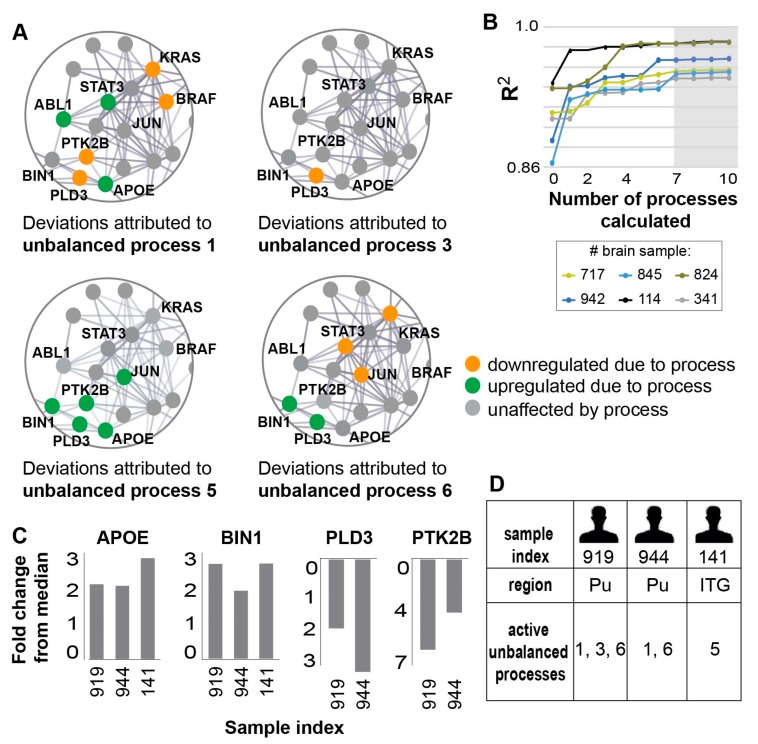
Seven unbalanced processes were identified in the 951 samples tested, underlying the disparities in biomarker expression levels in different samples. (**A**) Zoomed in images of four of the seven unbalanced processes identified in the dataset: 1,3, 5, and 6. For each process, the deviations in expression levels of nine transcripts are shown, demonstrating how different unbalanced processes can affect the expression levels of the same transcripts, sometimes in opposite directions. The complete information regarding the seven unbalanced processes and the transcripts affected by them can be found in Appendix A. (**B**) R^2^ values of selected samples were calculated by plotting the natural logarithm of the experimental data $\ln\left(X_{i}\left(k\right)\right)$ vs. $\sum G_{i\alpha }\lambda_{\alpha }(k)$ for different values of α. The value of R^2^ approaches 1 as more unbalanced processes are added to the calculation. Mathematically, 951 unbalanced processes are calculated for each patient. However, not all of them are significant. The figure shows that the R^2^ plots representing different patients reach a plateau after seven processes, suggesting that the first seven unbalanced processes are necessary to reproduce the experimental data, while the rest of the processes represent random noise in the system. Six selected brain samples are shown. The gray box highlights that the addition of the unbalanced process α>7 had no significant effect on the R^2^ value for these samples. (**C**) Fold change in expression level of the four AD biomarkers in three selected samples—the selected samples demonstrate similar deviations in expression levels of these biomarkers. (**D**) The samples were found to harbor different sets of active unbalanced processes. Hence, similar transcript expression levels can arise from distinct altered transcriptional signatures, possibly demanding different modes of treatment as well.

**Figure 5 biomolecules-10-00503-f005:**
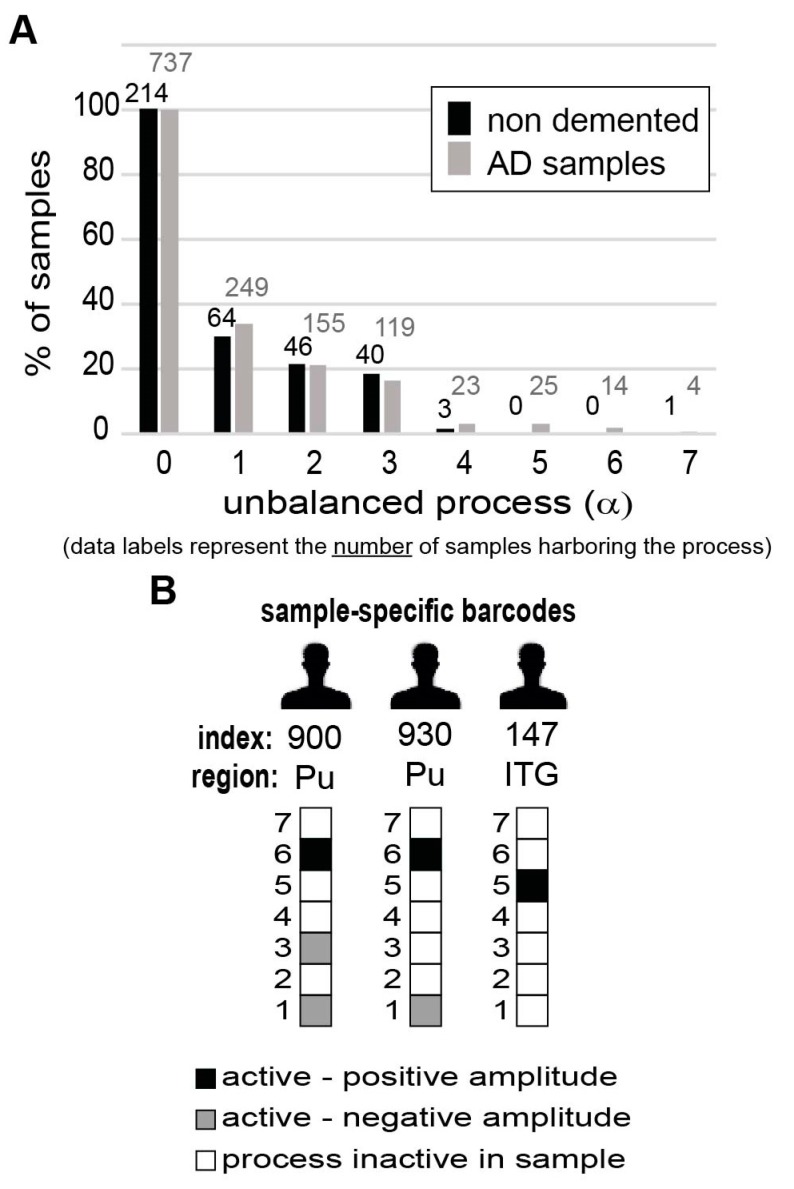
The altered transcriptional signature of every individual sample can be represented by a sample-specific barcode. (**A**) Frequency of expression of the different unbalanced processes in non-demented vs. AD samples. The graph shows that the seven unbalanced processes identified by surprisal analysis (SA) are each active in a similar percentage of non-demented vs. AD-bearing samples, and none can distinguish between the types of samples. (**B**) Sample-specific barcodes enable characterization of the complete altered transcriptional signature in every sample. The barcodes that characterize the three samples from Figure 4 are exemplified here. The sample-specific barcodes denote which unbalanced processes are active in the sample, including the sign of the amplitude.

**Figure 6 biomolecules-10-00503-f006:**
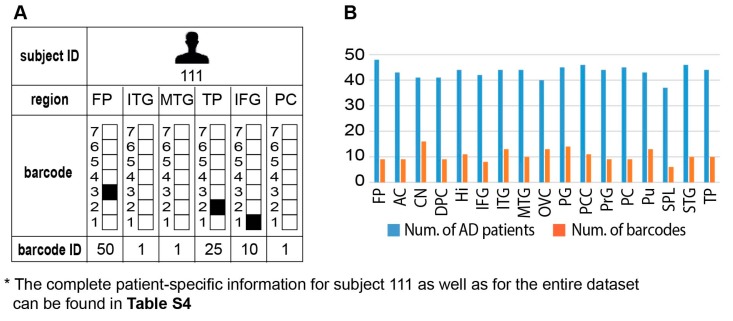
Intra-brain heterogeneity: different brain regions in the same subject can be characterized by distinct barcodes. (**A**) Subject 111 was selected to exemplify this point: four distinct barcodes were found to characterize six different brain regions in this subject. The complete information for subject 111 as well as for the entire dataset can be found in Appendix A. (**B**) In general, we found that every brain region was characterized by several distinct barcodes of unbalanced processes. The complete and detailed information can be found in Appendix A.

**Figure 7 biomolecules-10-00503-f007:**
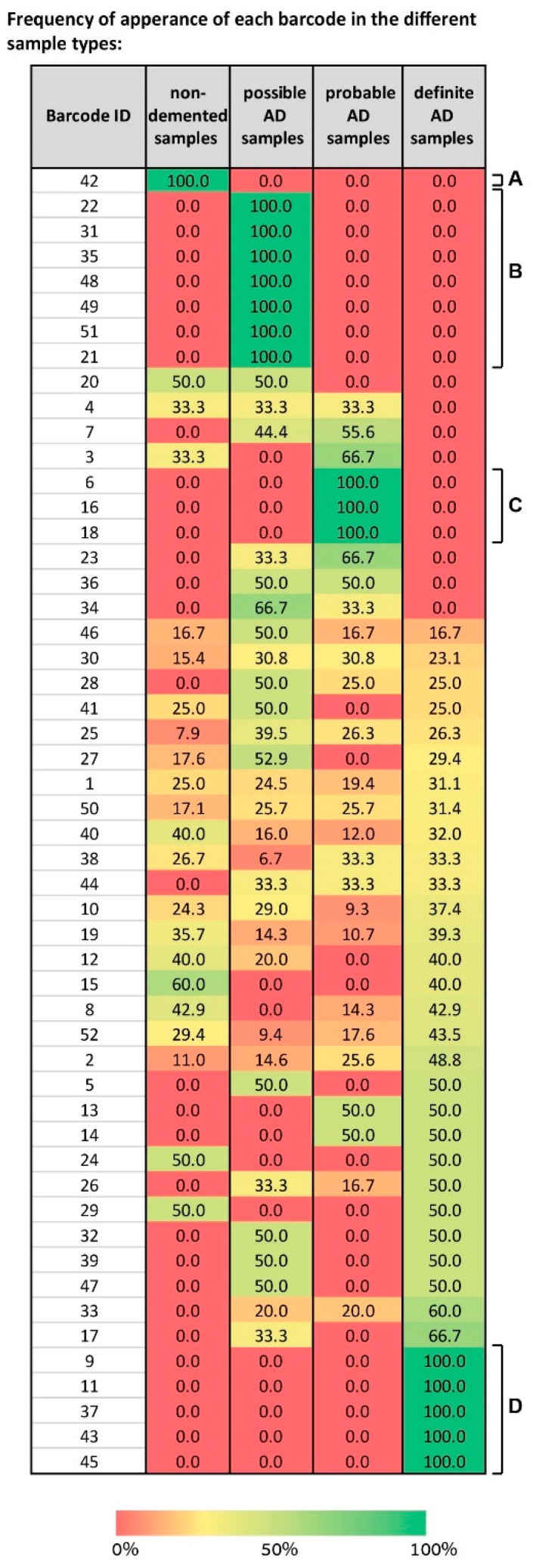
A spectrum of barcodes characterizes the 951 samples in the dataset, suggesting that we shift our conception of the molecular events in AD. The table presents the 52 barcodes identified in the dataset of 951 samples. For each barcode its frequency of expression in every sample type is shown. For example, blocks of barcodes characterizing solely one type of sample are evident (A, B, C, and D). However, the majority of the barcodes each characterize more than one type of sample, in different sub-combinations, suggesting that our conception of AD pathology should be shifted from biomarker-based characterization to multi-modal characterization, integrating, for example, biochemical and behavioral examinations.

## Data Availability

The data that support the findings of this study are available from [12] and GEO database (GSE1297).

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
