# Peer review of "Exploring Alzheimer’s Disease Molecular Variability via Calculation of Personalized Transcriptional Signatures"

_biomolecules, 2020, doi:10.3390/biom10040503_

Round 1

Reviewer 1 Report

The authors study the molecular heterogeneity of Alzheimer’s disease (AD) using an information-theoretic approach. They studied the transcriptional signature from a total of 951 brain samples obtained from 17 brain regions of 85 AD patients and 22 non-demented patients. The results showed that 30 different altered transcriptional signatures were characteristic of AD in comparison to non-demented subjects. Besides, the rest of the transcriptional signatures demonstrated a high complexity and variability of gene expression from the 951 samples studied, supporting the existence of multiple mechanisms implicated in the onset and development of this syndrome, and its overlap with the aging process.

The article presents exciting results, which may have implications in a more accurate diagnosis of the disease as well as the development of new therapeutic approaches. Therefore, the study needs to be published. There are a few comments that might be of interest.

1.- Since the initiation and progression of AD seem to be different in diverse brain regions, Is it possible that the transcriptional signatures found in the different brain regions of Alzheimer's patients are an expression of the different evolutionary stages of the disease rather than a heterogeneity of the syndrome?

2.- Is there any transcriptional signature or barcode associated with the aging process itself?

3.- Are the authors studying the possible correlation between the transcriptional patterns found and the expression of the proteins in the same regions assessed? Since there is a deregulated turnover of proteins in pathological conditions, proteomic studies potentially would reflect more accurately the molecular heterogeneity between different brain regions and patients.

4.- Is there a relationship between any transcriptional signature and the deregulation of the redox proteome in AD patients? And more specifically, with the cysteine proteome?

Author Response

Please note that the references to specific lines in the text consider that you are viewing the file either without tracked changes (i.e. the clean version) or the file with tracked changes in “simple markup” form.

1.- Since the initiation and progression of AD seem to be different in diverse brain regions, Is it possible that the transcriptional signatures found in the different brain regions of Alzheimer's patients are an expression of the different evolutionary stages of the disease rather than a heterogeneity of the syndrome?

It is a possibility that the transcriptional signatures in the different brain regions express different evolutionary stages of the disease. Our focus in this manuscript was to identify the sample-specific altered transcriptional signature at the time of sampling. This is because our interest was in highlighting the molecular differences between the samples, which may merit personalized treatment regimes.

The evolutionary aspect was out of the scope of our study and is an issue for further research.

For clarification, we have added the following paragraph to the discussion, in lines 636-642: “Accumulating data have shown that certain brain regions play important roles in AD pathology, and that the progression of the disease to different regions in the brain may characterize different stages of the disease [12,38,39]. However, our focus in this study was to identify the sample-specific altered transcriptional signature at the time of sampling. This is because our interest was in highlighting the molecular differences between the samples, which may merit personalized treatment regimes. The evolutionary aspect was out of the scope of our study and is an issue for further research.”

  1. Is there any transcriptional signature or barcode associated with the aging process itself?

We tried searching the unbalanced processes for aging-related processes. We failed to find a specific unbalanced process that is clearly related to aging. The aging processes seem to be intertwined with the AD processes, possibly because aging and neurodegeneration display common characteristics. For example, Gene Ontology (David) analysis showed that categories associated with memory and learning, appeared in unbalanced processes 1, 2, and 4 (See Supp. Table 2).

3.- Are the authors studying the possible correlation between the transcriptional patterns found and the expression of the proteins in the same regions assessed? Since there is a deregulated turnover of proteins in pathological conditions, proteomic studies potentially would reflect more accurately the molecular heterogeneity between different brain regions and patients.

We thank the reviewer for this well-placed comment. We agree that since proteins are the cellular entities that carry out processes, and since proteins are the targets for drugs, that it will be highly relevant to conduct a similar study of altered protein networks in AD samples. In this manuscript we selected a transcriptional dataset as proof of concept for our approach. The advantage of transcriptional datasets is that they usually consist of a large number of transcripts (~20-40 thousand), allowing for a higher-resolution analysis with less biological noise. Another important advantage of this particular dataset is a large number of samples (951) which significantly enhances the accuracy of the results as well. However, our data show that transcriptomic evaluation of AD samples may not suffice to uncover the complete set of altered molecular events in AD pathology. This is another important message we wished to convey in our work. We plan to analyze a large AD proteomeic dataset once it is available.

We have revised the Discussion to clarify this point. See lines 606-612: “ The finding that ~40% of the samples were not characterized by any of the unbalanced processes is interesting, because it suggests that transcriptomic evaluation of AD samples may not suffice to uncover the molecular processes gone awry in all cases of AD. In the future, when large datasets consisting of multiple samples, profiled simultaneously by transcriptomics, proteomics, and genomics, will be available, it will be important to integrate the transcriptional signatures with the proteomic signatures in AD, potentially allowing higher accuracy in the diagnosis, more precise categorization and personalized treatment of AD patients.”

4.- Is there a relationship between any transcriptional signature and the deregulation of the redox proteome in AD patients? And more specifically, with the cysteine proteome?

We have not found any significant pathway associated with the cysteine proteome in our Gene Ontology analysis (Supp. Table 2).

Since NFkB and its activation by ROS and proinflammatory cytokines have been highlighted in the pathogenesis of AD (Upinder Kaur et al., Reactive oxygen species, redox signaling and neuroinflammation in Alzheimer's disease: the NF-κB connection. Curr Top Med Chem 2015;15(5) 446-457. https://pubmed.ncbi.nlm.nih.gov/25620241/), we searched Supplementary Table 2 for NFkB and NFkB-related transcripts. We found that NFKBIA (a transcript encoding an inhibitor of NFkB) participates in processes 1, 4, 6, and 7. NFKBIZ (another transcript encoding an inhibitor of NFkB) participates in process 6. On the other hand, NFKB1, NFKB2, NFKBIB, NFKBID, NFKBIE and NFKBIL1 do not participate in any of the unbalanced processes.

Reviewer 2 Report

Reviewer’s comments to the authors

The authors employ an information-theoretic approach to uncover the molecular heterogeneity among a large cohort of Alzheimer’s disease (AD) and non-demented brain samples. The objective of the study is to obtain a better classification of AD patients into different categories. This sub-classification could benefit for the development of personalized therapies. The study is relevant, the methodology used (surprisal analysis) is really interesting and implies a novel viewpoint of AD. Also, the results obtained and supplementary material could be useful for the scientific community. Therefore, I encourage the publication. However, some issues should be addressed before. I have some minor concerns about the manuscript:

  1. The sample size is very large. However, the number of AD patients (85) and controls (22) is very different. It should be commented somewhere in which extent this difference could affect or not the results.
  2. Lines 57-59: The reference is missing in the sentence: “In addition, a comparison of Aβ production levels in brains of individuals who either suffered 58 from fAD, sporadic AD (sAD) or were not demented, indicated that sAD patients and non-demented 59 individuals show no significant differences in Aβ production levels.”
  3. 107-108: All AD samples are from sporadic AD patients and none of familiar AD. Why? When reading the text, an explanation for this decision is expected.
  4. Lines 128-138 and 226-234 are equal. Please suppress the information in results.
  5. Please check abbreviations (some are indicated several times, for example “caudate nucleus (CN)”)
  6. Some figures could be better shown. Figure 1 is excellent and really clarifying.
  7. Figure 2, however, is not clear to me. If the vertical axis adapted from 400-600, I think that two different levels could be discerned. For example, AC, TO, IFG nd CN seem to have a higher level. Is this relevant? What does it imply? Please indicate it in the legend to clarify.
  8. Figure 4 is very nice, too. Please use similar fonts for the labels, some are elongated (i.e. sample index) and others are to short (Sum of processes))
  9. Some comments made across the Results section could be better place into the Discussion. For example, lines 309-311 and 315-318 or the interpretation of the genes involved in each process.
  10. 359-362 states that the expression in different samples correspond to different processes. However, the example given is confusing (samples 919, 944 and 141), given two of the samples correspond to the same anatomical region and share processes. This example could induce thinking that the differences found depend on the region more than on the pathological condition.
  11. 402-404. The role of MAPK in memory and learning should be mentioned
  12. 405-408. I assume that some of these transcripts are related to glial activity? If it is the case, on the light of the newly described glymphatic system, these alterations could be of pottential interest. According to this hypothesis, solute clearance is altered in many pathologies, including AD and Parkinson's disease and could lead to the accumulation of toxic metabolites. Glial cells which are involved in development are also involved in this system.
  13. 417-418. Do non-demented samples also correspond to neocortical areas? the Authors should clarify if the regions are similar or not.
  14. Why only the main processes, 1 and 2 are explained? According to Figure 5, processes 5 and 6 are specific to AD's samples, and process 7 is very rare (if present, the graph is not clear). Their potential relevance should be also described. Also in lines 439-442 it is stated that all processes appear in both AD and normal cases and observing Fig. 5A, processes 5 and 6 are specific to AD samples. The legend should indicate this if this is the case. How many samples exhibit process 7? According to the graph, it seems that is non-existent. If there is any simple, the vertical axis should be addapted to allow a better visualization.
  15. 434-437 Given that obtaining brain biopsies is an invasive technique and cognitive disorders could arise from the surgical process, it should be of interest finding related plasma or LCR processes. This possibility should be mentioned.
  16. The figures referred to individual patients help to understand how each patient exhibits different patterns. However, I strongly recommend the authors to include an additional table and perhaps a figure considering the frequencies of each process in each region and pathological condition. For example, seeing the data in Supp. Table 4, barcodes 2 and 26 only appear in the Caudate nucleus in AD patients and never in normal samples, while other barcodes appear in the same region both in AD and normal samples. Another example: barcode 52 in the dorsolateral prefrontal cortex only appear in AD patients. Another: barcode 2 in the parahippocampal gyrus only appear in AD patients.
  17. 495-496. This sentence deserved further explaination, given that only AD patients showed this pattern in the CN. Could it be a possible biomarker?
  18. 532-535. Maybe these two views are complementaries: the search for commonalities makes easier to find biomarkers of the disease. However, the proposed viewpoint could be better addressed to the therapy.
  19. Validation of the robustness of the analysis. Why this section is not at the beggining of the results?
  20. A more deep discussion based on the bibliography could improve the paper. Only 7 references are used in the Discussion. Many comments are made along the Results section which should be better placed in the Discussion.
  21. 589-590 “resembling intra-tumor heterogeneity in cancer” Bibliography is missing to support this.

Reviewer 3 Report

Dagan et al. analyzed a large set of postmortem brain samples (many different brain regions) from both late-onset Alzheimer´s disease patients and non-demented individuals. The authors applied an information-theoretic approach to study alteration in sample-specific brain transcriptional networks, revealing 7 distinct subnetworks of interest of which a subset of 1-3 subnetworks (unbalanced processes) characterized each sample. Additionally, the authors identified many transcriptional signatures that discriminated patient samples from non-demented brains. Despite sharing common signatures, however, the Alzheimer´s patients did not share changes in same unbalanced processes at individual donor level, highlighting the molecular heterogeneity and complex nature of the disease. These are important but not unexpected observations. The work was well conducted and analyzed, and the paper is well written. However, the study has not yielded much in the way of new mechanistic insight.

The authors should include a table with descriptive statistics related to the patient material.

Alzheimer´s disease is a highly progressive disease, thus the observed molecular heterogeneity at individual donor level my reflect different stages of the disease. The authors should discuss this possibility.

Despite the method description is clear and sound, the authors should add some more description on how the transcriptional networks were constructed and not only cite it.

It is an editorial decision whether a descriptive study of this kind is suitable for the journal.
